# Empagliflozin Relaxes Resistance Mesenteric Arteries by Stimulating Multiple Smooth Muscle Cell Voltage-Gated K^+^ (K_V_) Channels

**DOI:** 10.3390/ijms221910842

**Published:** 2021-10-07

**Authors:** Ahasanul Hasan, Raquibul Hasan

**Affiliations:** Department of Pharmaceutical Sciences, College of Pharmacy, Mercer University, 3001 Mercer University Drive, Atlanta, GA 30341, USA; ahasanul.hasan@live.mercer.edu

**Keywords:** empagliflozin, mesenteric arteries, smooth muscle cell, voltage-gated K^+^ channels, vasodilation

## Abstract

The antidiabetic drug empagliflozin is reported to produce a range of cardiovascular effects, including a reduction in systemic blood pressure. However, whether empagliflozin directly modulates the contractility of resistance-size mesenteric arteries remains unclear. Here, we sought to investigate if empagliflozin could relax resistance-size rat mesenteric arteries and the associated underlying molecular mechanisms. We found that acute empagliflozin application produces a concentration-dependent vasodilation in myogenic, depolarized and phenylephrine (PE)-preconstricted mesenteric arteries. Selective inhibition of smooth muscle cell voltage-gated K^+^ channels K_V_1.5 and K_V_7 abolished empagliflozin-induced vasodilation. In contrast, pharmacological inhibition of large-conductance Ca^2+^-activated K^+^ (BK_Ca_) channels and ATP-sensitive (K_ATP_) channels did not abolish vasodilation. Inhibition of the vasodilatory signaling axis involving endothelial nitric oxide (NO), smooth muscle cell soluble guanylyl cyclase (sGC) and protein kinase G (PKG) did not abolish empagliflozin-evoked vasodilation. Inhibition of the endothelium-derived vasodilatory molecule prostacyclin (PGI_2_) had no effect on the vasodilation. Consistently, empagliflozin-evoked vasodilation remained unaltered by endothelium denudation. Overall, our data suggest that empagliflozin stimulates smooth muscle cell K_V_ channels K_V_1.5 and K_V_7, resulting in vasodilation in resistance-size mesenteric arteries. This study demonstrates for the first time a novel mechanism whereby empagliflozin regulates arterial contractility, resulting in vasodilation. Due to known antihypertensive properties, treatment with empagliflozin may complement conventional antihypertensive therapy.

## 1. Introduction

Type 2 diabetes (T2D) and hypertension are two major and independent risk factors for the development of cardiovascular diseases including heart disease, stroke, peripheral arterial disease and chronic kidney disease [1]. Hypertension and T2D often coexist, which multiplies the risk of adverse cardiovascular events in affected individuals [1,2]. Approximately 20% of patients with hypertension also have T2D, and 50% of patients with T2D have hypertension [3].

Clinical studies suggest that there is considerable residual risk for cardiovascular events in the hypertensive-diabetic group despite standard pharmacologic treatment and lifestyle modifications [4]. Therefore, there is a growing interest among clinicians with respect to management strategies for blood pressure lowering in this high-risk group that complement conventional antihypertensive therapy. This is based on a number of rigorous clinical trials that have shown that new hypoglycemic agents such as SGLT2 (sodium-glucose cotransporter-2) inhibitors can lower systemic blood pressure and have overall favorable cardiovascular outcomes [5,6,7,8,9,10,11,12,13,14,15,16,17]. In this regard, three members of SGLT2 inhibitors, namely empagliflozin, canagliflozin and dapagliflozin, have attracted much attention for use in the high-risk hypertensive-diabetic population due to their antihypertensive and hypoglycemic actions [18,19,20]. This blood pressure lowering effect is believed to be, at least in part, due to SGLT2 inhibitor-mediated vasodilation of systemic resistance arteries and a reduction in peripheral vascular resistance [5,21,22].

Empagliflozin is a potent and highly selective SGLT2 inhibitor that received U.S. FDA (The United States Food and Drug Administration) clearance for use in T2D in 2014 [23]. A significant body of evidence shows that empagliflozin treatment is associated with reduced arterial stiffness [24,25], vascular resistance [25] and blood pressure [17,26]. Recently, empagliflozin was reported to relax rabbit aorta by activating PKG and K_V_1.5 ion channels [27]. In addition, empagliflozin treatment was shown to decrease both systolic and diastolic pressure in rabbits [27]. In an earlier study, dapagliflozin was also shown to cause aorta relaxation by activating PKG and K_V_1.5 ion channels [21]. This raises the possibility that empagliflozin may also relax and reduce the resistance of systemic vessels such as small mesenteric arteries. Moreover, empagliflozin and other SGTL2 inhibitors were reported to possess significant antioxidant and anti-inflammatory actions that afford protection against nitro-oxidative stress and suppress vascular inflammation and atherosclerosis [5,28,29,30,31,32,33,34,35]. In this study, we investigated vasodilatory action of acute empagliflozin application in freshly isolated, resistance-size mesenteric arteries by using pressure myography and pharmacological approaches.

Our data demonstrate that empagliflozin stimulates smooth muscle cell voltage-gated K^+^ channels K_V_1.5 and K_V_7 to elicit vasodilation in resistance-size mesenteric arteries. This study is significant as it unveils a direct role for empagliflozin in relaxing mesenteric arteries that reduces vascular resistance and may explain blood pressure lowering effects of this drug. This study also lends additional support for the use of this drug in hypertensive-diabetic patients for reducing the risk of adverse cardiovascular events such as heart attack and stroke.

## 2. Results

### 2.1. Acute Empagliflozin Application Produces Vasodilation in Resistance Mesenteric Arteries

To examine the direct effect of acute empagliflozin treatment on arterial contractility, we performed pressure myography on small resistance-size mesenteric arteries. Freshly isolated third to fourth order mesenteric artery segments were cannulated and maintained in a temperature-controlled perfusion chamber continuously perfused with 37 °C PSS. Intravascular pressure gradually increased from 10 to 80 mmHg to stimulate the development of myogenic tone (~25% at 80 mmHg), and a cumulative concentration response to empagliflozin was performed. Our myography data show that empagliflozin produced a concentration-dependent vasodilation in mesenteric arteries (Figure 1A,B). Empagliflozin at 100 µM concentration produced a maximum dilation in mesenteric arteries by 13.31 ± 1.49% of passive arterial diameter at 80 mmHg (Figure 1C). Having examined the vasodilatory effect of empagliflozin in myogenic arteries, we next verified this finding in arteries pre-constricted with either 1 µM phenylephrine (PE) or 30 mM K^+^-PSS (30K), which depolarizes arteries to the same extent as 80 mmHg intraluminal pressure. We found that empagliflozin relaxed both PE and 30K-preconstricted arteries in a concentration-dependent manner (Figure 2A–D), similar to what was observed in pressurized myogenic arteries. Overall, these data demonstrate that empagliflozin is a vasodilator in resistance-size rat mesenteric arteries.

### 2.2. Empagliflozin-Induced Mesenteric Artery Vasodilation Is Independent of NO-sGC-PKG Signaling Axis

Endothelial cells (ECs) that line the lumen of all blood vessels are a critical regulator of smooth muscle contractility and vessel diameter. ECs produce vasodilators that act on underlying smooth muscle cells to elicit arterial vasodilation. NO, produced by eNOS, is one of the most prominent vasorelaxant factors that diffuses into arterial smooth muscle cells to stimulate sGC and cGMP production. Elevated cGMP levels activate PKG, which in turn activates myosin light chain phosphatase, resulting in vasodilation [36].

Considering the critical role of the NO-sGC-PKG signaling axis in vasodilation, we examined whether pharmacological inhibitors of this signaling axis could attenuate empagliflozin-mediated arterial vasodilation. We found that L-NNA, an inhibitor of eNOS, ODQ, a sGC inhibitor or KT5823, an inhibitor of PKG did not abolish empagliflozin-induced vasodilation (Figure 3A,B). These data suggest that the NO-sGC-PKG signaling axis is not involved in mesenteric artery vasodilation produced by empagliflozin.

### 2.3. Empagliflozin-Evoked Mesenteric Artery Vasodilation Does Not Depend on Endothelial PGI_2_

Cyclooxygenase (COX) enzymes in the endothelial cells synthesize another major vasodilator PGI_2_ that acts on its smooth muscle cell receptors to produce vasorelaxation [37,38]. Therefore, we asked if empagliflozin could stimulate PGI_2_ production to induce vasodilation. To perform this, we preincubated mesenteric arteries with indomethacin, an inhibitor of COX, and then applied empagliflozin. Our pressure myography data revealed that indomethacin application did not abrogate empagliflozin-induced vasodilation in mesenteric arteries (Figure 4A,B). These data suggest that empagliflozin-evoked vasodilation is not mediated by COX stimulation and PGI_2_ production.

### 2.4. Endothelium Denudation Does Not Alter Empagliflozin-Evoked Vasodilation

To assess the role of endothelium further, we compared empagliflozin-induced vasodilation in endothelium-intact and endothelium-denuded mesenteric arteries. Endothelium denudation was performed by slowly passing air bubbles through the vessel lumen and confirmed as described by others previously [39,40,41]. Briefly, we applied 1 µM acetylcholine (ACh) to endothelium-intact and endothelium-denuded arterial segments that had been preconstricted with 1 µM PE. We found that ACh fully reversed PE constriction in endothelium-intact arteries but not in endothelium-denuded arteries (Figure 5A,B). The application of 1 µM sodium nitroprusside (SNP), a NO donor, completely reversed PE constriction in both endothelium-intact and endothelium-denuded arteries (Figure 5A,B), suggesting that endothelium denudation does not affect smooth muscle responses to NO. Since ACh-induced vasodilation is primarily mediated by endothelial NO production, selective loss of ACh-evoked vasodilation demonstrates successful endothelium denudation [39,40,41]. We then compared empagliflozin responses in endothelium-intact and endothelium-denuded arteries. We found that empagliflozin treatment reversed PE-induced vasoconstriction both in endothelium-intact and endothelium-denuded arteries (intact 100% vs. denuded 101.24%) (Figure 5C,D). These data suggest that empagliflozin-elicited mesenteric artery vasodilation does not depend on endothelial signaling and likely involves smooth muscle.

### 2.5. Role of Smooth Muscle Cell Voltage-Gated K^+^ (K_V_) Channels in Empagliflozin-Induced Vasodilation

Arterial smooth muscle cells express a wide variety of K^+^ channels that control membrane potential and arterial contractility [42]. Several voltage-gated K^+^ channels (K_V_) are reported to be expressed in smooth muscle cells, and their activation causes K^+^ efflux, smooth muscle hyperpolarization and vasodilation [43,44].

Therefore, we sought to examine the hypothesis that empagliflozin may directly stimulate K_V_ channels to cause vasodilation in mesenteric arteries. Indeed, our data showed that the application of 4-aminopyridine (4-AP), a non-selective K_V_ blocker, [37,43,45] reduced empagliflozin-evoked vasodilation by ~60% (Figure 6A,B). These data suggest that K_V_ channel stimulation is critical for empagliflozin-mediated vasodilation in mesenteric arteries. We next explored the contribution of major K_V_ channel isoforms present in mesenteric artery smooth muscle cells, including K_V_1.3, K_V_1.5 and K_V_7 [43,44]. Our pressure myography data revealed that the application of DPO-1, a selective K_V_1.5 inhibitor [46,47], reduced empagliflozin response by ~27%. Interestingly, the application of linopirdine, a blocker of K_V_7 [48,49], suppressed empagliflozin-induced vasodilation by ~42%. When K_V_1.5 and K_V_7 channels were simultaneously inhibited by the co-application of DPO-1 and linopirdine, empagliflozin-evoked vasodilation was strongly suppressed by ~69% (Figure 6C,D), similar to 4-AP-meidated reduction (60%) of empagliflozin response (Figure 6A,B). The selective inhibition of K_V_1.3 channels by psora-4 did not suppress empagliflozin-induced vasodilation (Figure 6C,D). These findings provide compelling pharmacological evidence for the involvement of both K_V_1.5 and K_V_7 channels in mediating empagliflozin-induced vasodilation in small mesenteric arteries.

### 2.6. Role of Smooth Muscle Cell BK_Ca_ and K_ATP_ Channels in Empagliflozin-Induced Vasodilation

Arterial smooth muscle cells also express large-conductance Ca^2+^-activated K^+^ channels (BK_Ca_) and ATP-sensitive K^+^ channels (K_ATP_) which, when stimulated, caused arterial hyperpolarization and vasodilation [50].

Here, we analyzed the contribution of BK_Ca_ and K_ATP_ channels in empagliflozin-induced vasodilation. Myography data showed that the application of paxilline, an inhibitor of BK_Ca_ channels [51] and glibenclamide, a blocker of K_ATP_ channels [52], did not attenuate empagliflozin-induced vasodilation in resistance-size mesenteric arteries (Figure 7A,B). These data suggest that empagliflozin-evoked vasodilation is mediated by K_V_ channels, specifically K_V_1.5 and K_V_7 but not BK_Ca_ or K_ATP_ channels.

## 3. Discussion

In this study, we demonstrated for the first time that empagliflozin stimulates vasodilation in resistance arteries such as the small mesenteric arteries. Our data also demonstrate that empagliflozin-evoked vasodilation does not depend on endothelial signaling; rather, it depends on the activation of arterial smooth muscle cell K_V_1.5 and K_V_7 ion channels.

Accumulating evidence suggests that empagliflozin treatment is associated with substantial reduction in adverse cardiovascular events, such as heart attack and stroke, in diabetic patients [17]. Cardiovascular risk reduction has also been attributed to a decrease in arterial stiffness [24,25], contractility [25], vascular oxidative stress and inflammation in addition to improved cardiac function [5,31,33,34,35]. Previous studies have also shown that empagliflozin treatment lowers systemic blood pressure [17,26,27], resulting in better cardiovascular outcomes [17]. Systemic blood pressure is a function of peripheral vascular resistance exerted by small arteries and arterioles. It has been proposed that empagliflozin may lower blood pressure by reducing vascular resistance [27]. However, experimental evidence demonstrating a reduction in vascular resistance is lacking. Our study demonstrates for the first time that empagliflozin produces vasodilation in resistance-size mesenteric arteries which are part of the systemic vasculature and contribute to total peripheral resistance. A recent study by Seo et al. (2020) showed that empagliflozin treatment reduces the reactivity of rabbit thoracic aorta rings [27]. Since the aorta is a conduit vessel that is primarily involved in pressure damping and distribution of blood to the periphery without any major contribution to the peripheral vascular resistance, the relevance of such findings in systemic blood pressure regulation could not be ascertained. Our study establishes a correlation between empagliflozin-induced blood pressure lowering and a reduction in vascular resistance through vasodilation of resistance mesenteric arteries. Mesenteric arteries which are part of splanchnic circulation receive over 25% of cardiac output [53]. Therefore, the relaxation of small mesenteric arteries may contribute appreciably to the reduction in peripheral vascular resistance and systemic blood pressure. However, further studies will be required to assess if acute empagliflozin treatment could also lower systemic blood pressure.

Here, we demonstrated that empagliflozin relaxes myogenic (80 mmHg-constricted), 30K- and PE-constricted resistance-size mesenteric arteries (Figure 1 and Figure 2). Intraluminal pressure of 80 mmHg or 30K^+^ PSS opens voltage-gated Ca^2+^ channel Ca_V_1.2 in arterial smooth muscle cells, resulting in Ca^2+^ influx, rise of intracellular Ca^2+^ concentration and vasoconstriction. PE binds to smooth muscle cell α1 adrenergic receptor (G_q11_-coupled), which activates phospholipase C (PLC), resulting in the formation of inositol triphosphate (IP_3_). IP_3_ binds to ryanodine receptors on sarcoplasmic reticulum (SR) to stimulate the release of stored SR Ca^2+^ into the cytoplasm, causing smooth muscle contraction [54]. Although these pathways tend to overlap and activate each other, the efficacy of empagliflozin in relaxing mesenteric arteries regardless of the stimuli reinforces our finding that empagliflozin is a potent vasodilator with the ability to relax arteries preconstricted with different vasoconstrictors. Seo et al. (2020) showed that empagliflozin relaxes rabbit aorta by activating PKG and K_V_1.5 channels [27]. Our findings are partly in agreement with this study in that empagliflozin stimulates K_V_1.5 channels in mesenteric arteries, as it did in the aorta. Furthermore, our study demonstrates that empagliflozin-induced vasodilation in mesenteric arteries is dependent on the activation of two smooth muscle cell K_V_ ion channels, K_V_1.5 and K_V_7 (Figure 6). We demonstrated that, in mesenteric arteries, K_V_7 inhibition produces 15% more vasodilation than the inhibition of K_V_1.5 channels. As K_V_1.5 channels are the predominant K_V_ isoform (65%) [41,44] in mesenteric arteries, the involvement of K_V_7 which has relatively lower abundance (<10% of all K_V_) [41] may indicate a greater selectivity of empagliflozin for K_V_7 over K_V_1.5 channel. This suggests that K_V_7 channel activation is the major contributor to empagliflozin-elicited vasodilation in mesenteric arteries. In contrast, a lack of involvement of K_V_7 channels in mediating empagliflozin-induced relaxation of the aorta may be attributed to the difference between aorta and mesenteric artery microenvironment. Unlike the previous finding that suggests a role for PKG in aorta relaxation, our data rule out the contribution of PKG activation and its upstream signaling components involving sGC and endothelium-derived NO in empagliflozin-induced mesenteric artery vasodilation. These findings suggest that, although PKG activation is required for empagliflozin-induced relaxation of aorta, it has no role in mesenteric artery vasodilation or the activation of K_V_1.5 and K_V_7 channels. Since empagliflozin relaxed 30K-constricted mesenteric arteries, it is possible that other mechanisms, in addition to K_V_ ion channels, are also activated by this drug to induce vasodilation. Regardless of the mechanisms involved, relaxation of resistance mesenteric arteries upon acute empagliflozin application may be beneficial for reducing blood pressure and the risks of heart attack and stroke that are associated with elevated arterial tone and hypertension.

SGLT2 inhibitors including empagliflozin have been shown to have a wide range of effects in the cardiovascular system and beyond [5,28,29,30,31,32,33,34,35] Although they are primarily recommended for decreasing renal glucose reabsorption by selectively blocking SGLT2 [55], this class of drugs appears to have many other molecular targets that result in a range of glucose independent beneficial effects or ‘pleiotropic’ effects. Since SGTL2 has the highest concentration in the renal proximal tubule, the vasodilatory action of empagliflozin in mesenteric artery may be another example of a pleiotropic effect of SGLT2 inhibitors. Future studies are likely to identify more targets of empagliflozin action in the cardiovascular and other systems that may explain improved cardiovascular function in diabetic patients.

Our study has several limitations. Since our findings are based on healthy, young animals, future studies using diabetic animals will be required to validate the vasodilatory action of this drug in a relevant model. We tested the vasodilatory action of empagliflozin in an ex vivo preparation of resistance-size mesenteric arteries. Proof of concept studies such as blood pressure measurement in ambulatory animals as well as in vivo blood flow measurement will be required to understand the full therapeutic potential of this drug in blood pressure and blood flow regulation. Similar experiments in human vessels and the measurement of blood flow and blood pressure will significantly enhance the translational aspect of this work.

In summary, we demonstrated that empagliflozin relaxes resistance-size mesenteric arteries by stimulating smooth muscle cell K_V_1.5 and K_V_7 ion channels. Empagliflozin-evoked vasodilation does not depend on endothelial signaling or other smooth muscle K^+^ channels. This finding is significant as it uncovers a direct role for empagliflozin in reducing vascular resistance, which may provide an explanation for its blood pressure lowering action in clinical studies. As diabetes and hypertension coexist, the use of empagliflozin (perhaps other SGLT2 inhibitors) in this high-risk diabetic population may complement conventional antihypertensive therapy to potentially reduce the overall risk of adverse cardiovascular events such as heart attack and stroke.

## 4. Materials and Methods

### 4.1. Animals

All animal protocols were approved by the Mercer University Institutional Animal Care and Use Committee (IACUC, Ref. No. A2107011, last approved on 11 August 2021) Male Sprague Dawley (SD) [35] rats, 8–9 weeks of age, were used for this study. Animals were purchased from Charles River Laboratories (Wilmington, MA, USA) and acclimatized in the vivarium of the Mercer University College of Pharmacy for one week before experimentation. Rats were individually caged in a temperature-regulated room (temperature 22 ± 2 °C; 55% humidity; and 12 h light/dark cycles) and had access to standard chow diet and drinking water ad libitum. Animals were randomized for all experiments. The experiments were conducted in accordance with the guidelines set by the United States National Institutes of Health [56] Guide for the Care and Use of Laboratory Animals.

### 4.2. Tissue Preparation

Animals were euthanized by passing compressed CO_2_ gas into a CO_2_ euthanasia chamber followed by decapitation. An entire mesenteric artery bed was dissected and placed into ice-cold PSS (see composition below). Third and fourth order branches of mesenteric arteries (150–250 µm) were dissected cleaned of adventitial tissue in ice-cold PSS. Cleaned arterial branches were cut into 1–2 mm long segments that were kept in ice-cold PSS until cannulated for pressure myography [39,40,41,57].

### 4.3. Solutions and Chemicals

PSS for vessel dissection, isolation and preparation contained (in mM) the following: KCl 6.0, NaCl 112, NaHCO_3_ 1.18, MgSO_4_ 1.18, KH_2_PO_4_ 1.18, CaCl_2_ 1.8 and glucose 10. The pH of the PSS was adjusted to and maintained at 7.4 by continuous flow of normal air into the PSS. The amount of 60 mM K^+^-PSS (60K) was prepared by equimolar replacement of NaCl with KCl in the PSS. Empagliflozin was purchased from Ambeed Inc. (Arlington Heights, IL, USA) and used in the concentration range of 0.001–100 µM. Phenylephrine (PE) and 4-aminopyridine (4-AP) were purchased from Sigma-Aldrich (St. Louis, MO, USA) and used at a final concentration of 1 µM and 1 mM, respectively. Indomethacin (10 µM), DPO-1 (1 µM), Linopirdine (10 µM), Psora-4 (100 nM), Glibenclamide (10 µM), paxilline (10 µM), ODQ (10 µM), KT 5823 (1 µM), L-NNA (10 µM), SNP (10 µM) and acetylcholine (ACh, 1 µM)) were purchased from Tocris (Minneapolis, MN, USA). 4-AP, SNP and PE were dissolved in distilled water. Empagliflozin, DPO-1, linopirdine, psora-4, indomethacin, glibenclamide, paxilline, ODQ, KT5823, L-NNA and ACh were dissolved in dimethyl sulfoxide (DMSO). The final DMSO concentration (<0.1%) in the myograph chamber had no significant effect on arterial contractility.

### 4.4. Pressure Myography

Arterial segments were cannulated in a perfusion chamber (Living Systems Instrumentation, St. Albans, VT, USA), continuously perfused with 37 °C PSS and gassed with a mixture of 21% O_2_, 5% CO_2_ and 74% N_2_ to maintain pH 7.4. A 60K-induced constriction was used for testing the viability of the arteries at 10 mmHg before all experiments. Intraluminal pressure gradually increased to 80 mmHg with increments of 20 mmHg at each step to allow the development of myogenic tone. Intraluminal pressure was altered using a pressure servo controller with peristaltic pump (PS-200, Living Systems) and monitored with a pressure monitor. At 80 mmHg, mesenteric arteries developed 25 ± 3.91% myogenic tone, after which increasing concentrations of empagliflozin (0.001–100 µM) were applied. Diameter changes were recorded at 1 Hz using a charge-coupled device (CCD) camera connected to a Nikon Ts2 microscope and an automatic edge-detection function of IonWizard software (IonOptix, Milton, MA, USA). Luminal flow was absent during the experiment. Myogenic tone (%) was calculated as follows: 100 × (1 − D_active_/D_passive_), where D_active_ is active arterial diameter in the presence of Ca^2+^ and D_passive_ is the passive arterial diameter determined in Ca^2+^-free PSS supplemented with 5 mM EGTA [39,40,41,57].

For the mechanistic studies, we used 1 μM PE-preconstricted arteries. Arteries were preincubated for 15 min with different pharmacological modulators prior to the application of empagliflozin in the presence of the modulator.

### 4.5. Endothelium Denudation

Endothelium was denuded by slow passage of air bubbles through the vessel lumen for 1–2 min. Absence or at least 90% reduction in ACh-induced vasodilation was used as the indicator of successful endothelium denudation [39,40,41]. Sodium nitroprusside (SNP)-induced vasodilation and myogenic-induced and 60K-induced vasoconstriction remained unaltered by endothelium denudation.

### 4.6. Statistical Analysis

OriginLab software version 9.55 (2019b) (OriginLab, Northampton, MA, USA) was used for data analysis. The results were expressed as mean ± SEM. Considering the interval-ratio data with normal distribution, we performed unpaired Student’s *t*-tests (two-tailed) to test hypotheses. A *p* value of <0.05 was considered statistically significant [28,29,39].

## Figures and Tables

**Figure 1 ijms-22-10842-f001:**
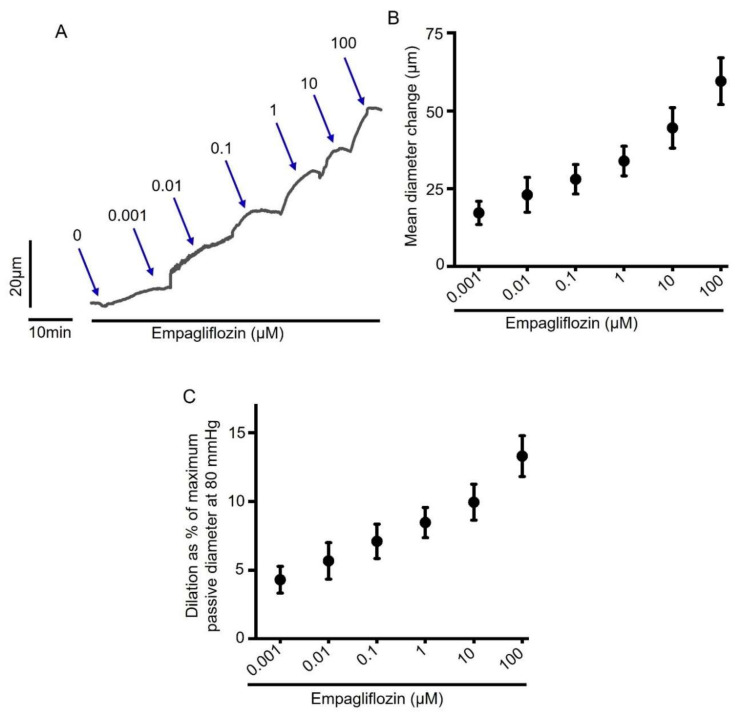
Empagliflozin stimulates concentration-dependent vasodilation in myogenic resistance-size mesenteric arteries. (**A**) An original trace illustrating vasodilation in a rat mesenteric artery at different concentrations (0.001 to 100 µM) of empagliflozin. (**B**) Mean data for empagliflozin-induced vasodilation, *n* = 6. (**C**) Mean data for empagliflozin-induced vasodilation expressed as % of maximum passive diameter at 80 mmHg, *n* = 6.

**Figure 2 ijms-22-10842-f002:**
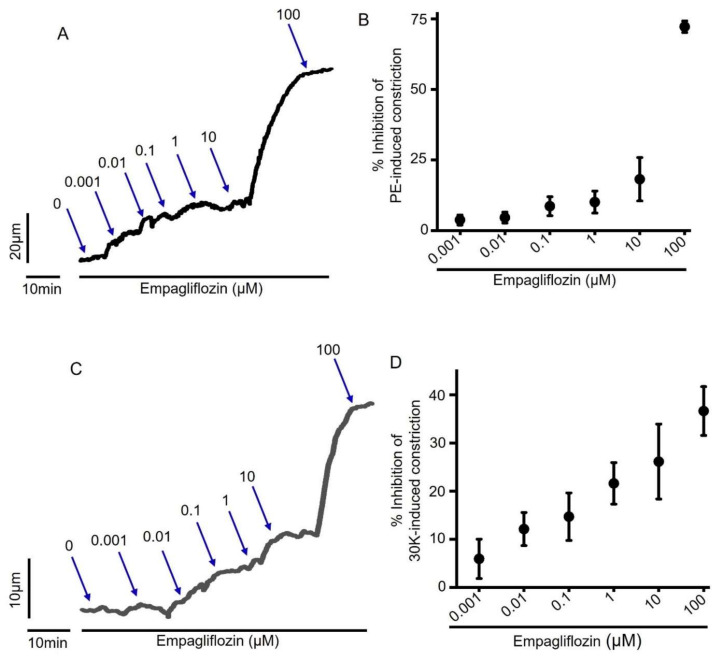
Empagliflozin stimulates vasodilation in PE- and 30K-constricted mesenteric arteries. (**A**) An original trace showing concentration-dependent inhibition of 1 µM PE-induced contraction in a mesenteric artery by empagliflozin (0.001 to 100 µM). (**B**) Mean data for % inhibition of PE-induced constriction by empagliflozin, *n* = 3. (**C**) An original trace showing concentration-dependent inhibition of 30K-induced constriction in a mesenteric artery by empagliflozin (0.001 to 100 µM). (**D**) Mean data for % inhibition of 30K-induced constriction by empagliflozin, *n* = 6.

**Figure 3 ijms-22-10842-f003:**
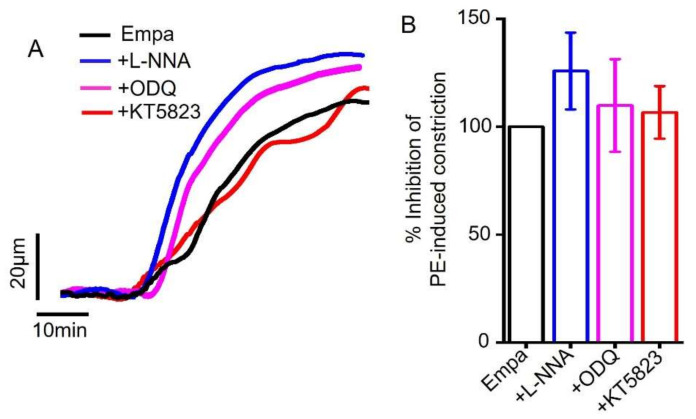
Empagliflozin-induced vasodilation in resistant mesenteric arteries is independent of NO-sGC-PKG signaling axis. (**A**) Original traces comparing empagliflozin (Empa)-induced vasodilation with or without pharmacological modulators (10 µM L-NNA, 10 µM ODQ and 1 µM KT5823) of NO-sGC-PKG signaling axis. (**B**) Mean data comparing Empa-induced mesenteric artery vasodilation with or without pharmacological inhibitors of NO-sGC-PKG signaling, *n* = 4.

**Figure 4 ijms-22-10842-f004:**
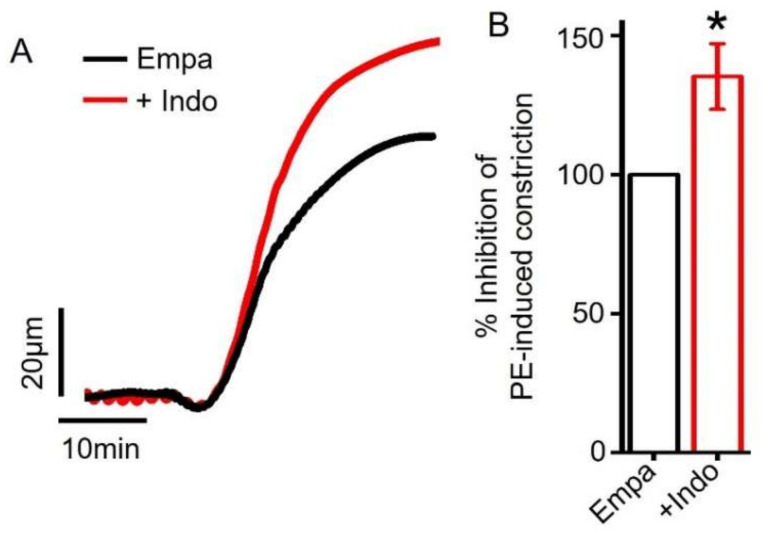
Empagliflozin-induced vasodilation is not mediated by endothelial PGI_2_ production. (**A**) Original traces comparing Empa-induced vasodilation with or without indomethacin (10 µM). (**B**) Mean data comparing Empa-induced mesenteric artery relaxation with or without indomethacin, *n* = 7. * *p* < 0.05 vs. Empa.

**Figure 5 ijms-22-10842-f005:**
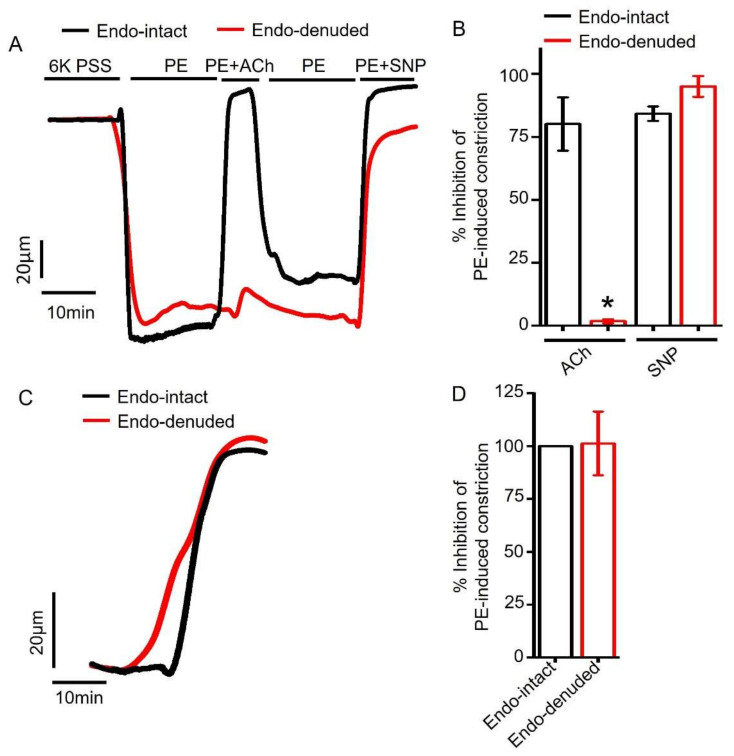
Empagliflozin-induced vasodilation is endothelium independent. (**A**) Original traces comparing responses of PE-constricted endothelium (endo) intact and denuded mesenteric arteries to ACh (1 µM) and SNP (10 µM). (**B**) Mean data showing selective loss of ACh-induced vasodilation but not SNP-elicited vasodilation in endo-denuded and PE-constricted arteries, *n* = 4. * *p* < 0.05 vs. endo-intact. (**C**) Original traces comparing Empa-induced (100 µM) vasodilation in endo-intact and endo-denuded arteries. (**D**) Mean data comparing Empa-induced vasodilation in endo-intact and endo-denuded mesenteric arteries, *n* = 4.

**Figure 6 ijms-22-10842-f006:**
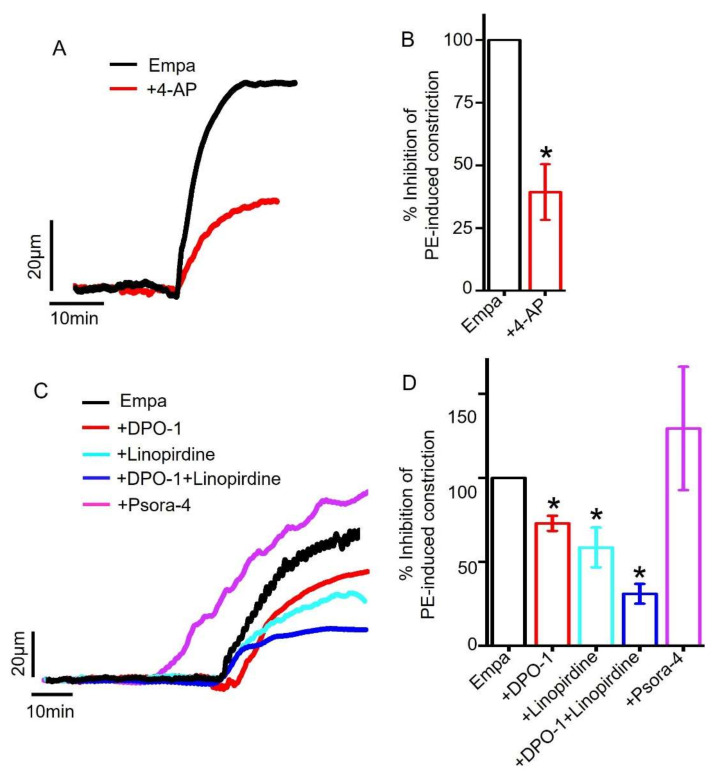
Role of smooth muscle cell K_V_ channels in empagliflozin-induced vasodilation. (**A**) Original traces showing that non-selective inhibition of K_V_ channels with 4-AP (1 mM) reduced Empa-induced vasodilation in mesenteric arteries. (**B**) Mean data illustrating significant reduction in Empa-evoked vasodilation by non-selective inhibition of K_V_ channels, *n* = 4. * *p* < 0.05 vs. Empa. (**C**) Original traces illustrating the effects of K_V_1.5, K_V_7 and K_V_1.3 channel inhibitors (1 µM DPO-1, 10 µM linopirdine and 100 nM psora-4, respectively) on Empa-induced vasodilation in resistance mesenteric arteries. (**D**) Mean data comparing the modulation of Empa-evoked vasodilation by KV1.5, KV7 and KV1.3 channel inhibition, *n* = 3–7. * *p* < 0.05 vs. Empa.

**Figure 7 ijms-22-10842-f007:**
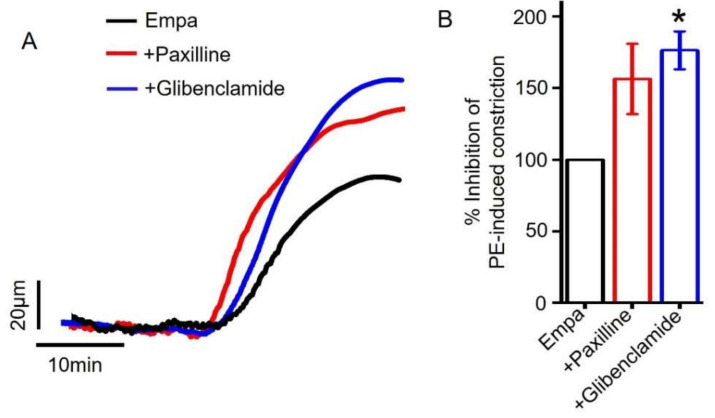
Role of smooth muscle cell BK_Ca_ and K_ATP_ channels in empagliflozin-induced vasodilation. (**A**) Original traces showing that the inhibition of BK_Ca_ and K_ATP_ channels (by 10 µM paxilline and 10 µM glibenclamide, respectively) does not reduce Empa-induced vasodilation in resistance-size mesenteric arteries. (**B**) Mean data showing that BK_Ca_ and K_ATP_ channel inhibition does not attenuate Empa-induced vasodilation, *n* = 3–5. * *p* < 0.05 vs. Empa.

## Data Availability

Data is available within the article.

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
