# Peer review of "Empagliflozin Relaxes Resistance Mesenteric Arteries by Stimulating Multiple Smooth Muscle Cell Voltage-Gated K+ (KV) Channels"

_ijms, 2021, doi:10.3390/ijms221910842_

Round 1
Reviewer 1 Report
The authors have done a nice study investigating the effects of Empagliflozin on rat mesenteric arteries and the mechanisms, which may further interpret the clinical hypotensive effect of Empagliflozin. My only concerns are:
1. The dosage used in the study seems to be higher than the maximum plasma concentration in human studies with normal dosage, is it applicable to clinical treatment? Please integrate clinical data with Empagliflozin metabolism and blood concentration in the discussion.
2. Empagliflozin is used on diabetic patients, for the animal study, the authors used relatively young normal rats, I suggest also using diabetic rats to observe the alterations of the resistance and K+ channels upon Empagliflozin treatment.
Author Response
Responses to the Editor and the Reviewers
We thank the editor and reviewers for their valuable comments towards the improvement of the manuscript. We have modified the manuscript in accordance with your suggestions and consider it significantly improved for publication.
Reviewer 1
The authors have done a nice study investigating the effects of Empagliflozin on rat mesenteric arteries and the mechanisms, which may further interpret the clinical hypotensive effect of Empagliflozin. My only concerns are:
Comment 1. The dosage used in the study seems to be higher than the maximum plasma concentration in human studies with normal dosage, is it applicable to clinical treatment? Please integrate clinical data with Empagliflozin metabolism and blood concentration in the discussion.
Response: We greatly appreciate your comment on the manuscript. We thank you for raising this point. Although we used 100 µM for the mechanistic studies, we observed concentration-dependent vasodilation in myogenic, PE- and 30K-constricted arteries. Empagliflozin is prescribed up to 100 mg/day. Empagliflozin is absorbed rapidly and reaches a peak plasma concentration of 2.75µΜ within 3 hours after administration of 100 mg (Heise et al., 2013). We agree that the concentration of empagliflozin used was higher than that used clinically but this is only for the mechanistic study. A lower empagliflozin concentration, 1μM that falls within the therapeutic range also evoked strong vasodilation when compared to 100µM response (34 µm Vs. 60 µm) (Fig. 1B). Therefore, 1μM and smaller concentrations are likely to produce vasodilation that could influence blood flow and pressure. Furthermore, overmedication or abuse of empagliflozin can raise the blood concentration of empagliflozin. Therefore, our results should be fully considered when prescribing empagliflozin to patients with cardiovascular diseases such as hypotension (Seo et. a. 2020).
Heise, T.; Seman, L., Macha, S.; Jones, P.; Marquart, A.; Pinnetti, S., Woerle, H. J.; Dugi, K. Safety, tolerability, pharmacokinetics, and pharmacodynamics of multiple rising doses of empagliflozin in patients with type 2 diabetes mellitus. Diabet. Ther. 2013, 4, 331–345. DOI: 10.1007/s13300-013-0030-2
Seo, M. S.; Jung, H. S.; An, J. R.; Kang, M.; Heo, R.; Li, H.; Han, E. T.; Yang, S. R.; Cho, E. H.; Bae, Y. M.; Park, W. S. Empagliflozin dilates the rabbit aorta by activating PKG and voltage-dependent K(+) channels. Toxicol Appl Pharmacol. 2020, 403, 115153. DOI: 10.1016/j.taap.2020.115153
Comment 2. Empagliflozin is used on diabetic patients, for the animal study, the authors used relatively young normal rats, I suggest also using diabetic rats to observe the alterations of the resistance and K+ channels upon Empagliflozin treatment.
Response: Thank you for the suggestion. Having established the mechanism of empagliflozin-induced vasodilation in resistance mesenteric arteries from young healthy animals, our future studies will investigate if empagliflozin could relax mesenteric arteries from diabetic animals and reduce systemic blood pressure.
Reviewer 2 Report
Comment 1: In the result 2.1, more detailed explanation is needed. (The intraluminal pressure is increased from ? mmHg to 80 mmHg) And how much of the pressure they give as the resting pressure? (The author described that they tested viability at 10 mmHg and then the pressure is increased 80 mmHg with increment of 20 mmHg.)
Comment 2: Can author describe why they examine the effects of empagliflozin at 80 mmHg instead of at 40 mmHg (resting pressure)?
Comment 3: The abbreviation comes after the full name when it is first appeared.
Author Response
Responses to the Editor and the Reviewers
We thank the editor and all reviewers for their valuable comments towards the improvement of the manuscript. We have modified the manuscript in accordance with your suggestions and consider it significantly improved for publication.
Reviewer 2
Comment 1: In the result 2.1, more detailed explanation is needed. (The intraluminal pressure is increased from ? mmHg to 80 mmHg) And how much of the pressure they give as the resting pressure? (The author described that they tested viability at 10 mmHg and then the pressure is increased 80 mmHg with increment of 20 mmHg.)
Response: We thank the reviewer for this comment. After cannulation, pressure is increased from 0 to 10mmHg at which we applied 60K PSS to check the viability of the mounted vessel segments. Of note, 60K response at this pressure also gives an accurate measure of CaV1.2 channel function without interference from myogenic vasoconstriction that occurs at higher pressures. Intraluminal pressure is then increased to 20 mmHg, 40 mmHg, 60mm Hg, and 80 mmHg. Intraluminal pressure is maintained at least for 10-12 minutes to get a stable baseline at each pressure step.
Comment 2: Can author describe why they examine the effects of empagliflozin at 80 mmHg instead of at 40 mmHg (resting pressure)?
Response: Thank you for raising this interesting question. In this study, we used three models to examine the vasodilatory effects of empagliflozin. One of the models is the myogenic (80mmHg-constricted) mesenteric artery that had ~25% tone at 80 mmHg. Myogenic response is an important component of blood flow autoregulation that occurs in the pressure range of 70 and 175 mmHg. At 80mmHg, resistance mesenteric arteries developed ~25% myogenic tone due to depolarization of the smooth muscle cells. According to a previous study, active myogenic tone at 80 mmHg would be closer to 50% in the muscle arteries, and the predicted membrane potential would be near −37 mV (Kotecha et. al. 2005 and Dora et. al. 2005). Therefore, 80 mmHg intraluminal pressure was used as a reference physiological pressure to stimulate myogenic tone to study vasodilatory action of empagliflozin. However, we used 40mmHg pressure for 30K- and PE-preconstricted artery models, as myogenic tone was negligible at 40 mmHg and did not interfere with 30K- and PE-preconstricted artery models.
Kotecha, N.; Hill, M. A. Myogenic contraction in rat skeletal muscle arterioles: smooth muscle membrane potential and Ca2+ signaling. Am J Physiol Heart Circ Physiol. 2005, 289(4), H1326-34. DOI: 10.1152/ajpheart.00323.2005.
Dora, K. A. Does arterial myogenic tone determine blood flow distribution in vivo? Am J Physiol Heart Circ Physiol. 2005, 289(4), H1323-5. DOI: 10.1152/ajpheart.00513.2005.
Comment 3: The abbreviation comes after the full name when it is first appeared.
Response: Thank you for your suggestion. We have modified the text accordingly.
Reviewer 3 Report
This manuscript describes studies testing the hypothesis that empagliflozin activates KV channels to produce vasodilation. The major concern that I have relates to the authors finding that empagliflozin effectively dilated mesenteric resistance arteries contracted with 30 mM KCl solutions. This concentration of K+ will shift the Nernst potential (the equilibrium potential) for K+ from about -80 mV to about -30 to -40 mV and clamp the membrane potential at this potential. As such, opening of K+ channels will have no effect on membrane potential because the electrochemical gradient for K_ diffusion is eliminated (we routinely use this approach as a non-selective means to test for K+ channel activity of vasodilators and have verified our results with membrane potential measurements and patch clamp). Thus, in addition to producing vasodilation by activating KV channels, empagliflozin must have some additional mechanism of action. This must be adequately discussed in the Discussion.
Author Response
Responses to the Editor and the Reviewers
We thank the editor and reviewers for their valuable comments towards the improvement of the manuscript. We have modified the manuscript in accordance with your suggestions and consider it significantly improved for publication.
Reviewer 3
Comment: This manuscript describes studies testing the hypothesis that empagliflozin activates KV channels to produce vasodilation. The major concern that I have relates to the authors finding that empagliflozin effectively dilated mesenteric resistance arteries contracted with 30 mM KCl solutions. This concentration of K+ will shift the Nernst potential (the equilibrium potential) for K+ from about -80 mV to about -30 to -40 mV and clamp the membrane potential at this potential. As such, opening of K+ channels will have no effect on membrane potential because the electrochemical gradient for K_ diffusion is eliminated (we routinely use this approach as a non-selective means to test for K+ channel activity of vasodilators and have verified our results with membrane potential measurements and patch clamp). Thus, in addition to producing vasodilation by activating KV channels, empagliflozin must have some additional mechanism of action. This must be adequately discussed in the Discussion.
Response: Thank you for sharing this interesting perspective. We agree with the reviewer and have added a few lines in the discussion. However, the notion that opening of K+ channels will have no effect on membrane potential because the electrochemical gradient for K+ diffusion is eliminated, may not be always true. Previously, Marathan and co-authors (Marthan et. al., 1993) reported that cromakalim causes vasodilation through the activation of potassium (K+) channels. In that study, cromakalim (0.1-1 µM) was reported to inhibit KCl induced contraction. But the effect was shown to be limited to low KCl concentration (<40 mM) at which the membrane potential is shifted to approximately -40, -35 mV. These values are close to the threshold membrane potential for the activation of voltage-gated Ca2+ channels. Cromakalim hyperpolarizes the membrane by opening K+ channels, which will lead to the closure of voltage-gated Ca2+ channels. In contrast, cromakalim had no effect on contraction induced by 50-100 mM KCl concentrations that largely depolarize the membrane and reduce the driving force of K+ ions. In that study, cromakalim at higher concentrations (10µM) decreased the maximal KCl-induced contraction. Our data appears to be more consistent with this study. However, we agree with the reviewer that a non KV channel mechanism may also be involved.
Savineau, J. P.; Marthan, R. Effect of cromakalim on KCl-, noradrenaline- and angiotensin II-induced contractions in the rat pulmonary artery. Pulm Pharmacol. 1993, 6(1), 41-8. DOI: 10.1006/pulp.1993.1007.
Reviewer 4 Report
8.26.2021
This study by Ahasanul Hasan and Raquibul Hasan utilized Pressure Myography method and pharmacology tools to investigate the mechanism of Empagliflozin reduce systemic blood pressure. All experiments were well preformed, but the whole experimental design is not perfect! I have some recommendations to make this study strong.
1. Thinking about to do either one of the following experiments:
a. To get the proof of SGLT2 protein expressed in the vascular smooth muscle cells from rat mesenteric arteries (if the action of Empagliflozin on Kv channels through SGLT2);
b. To see the effects of Empagliflozin on Kv channels (patch clamp electrophysiology recording), utilizing exogenous expression system to over express Kv1.5 and Kv7 separately (if the action of Empagliflozin directly on Kv channels).
2. Please include original traces for Fig. 1C, Fig. 1F, and Fig. 2.
3. Please clearly label the Y axis unit (percentage or µm) for Fig. 1B
4. For the Fig. 3B, please explain why the vasodilation increased dramatically after co-application of EMPA and Indomethacin.
5. For the Fig. 6B, please explain why the vasodilation increased dramatically after co-application of EMPA and Paxiline, as well as EMPA and Glibenclamide.
6. For the experiment of Fig. 1A, I am wondering if the vasodilation response will be reversed when the 100 µM Empagliflozin was washed away.
7. I don’t know whether using such high concentration (100 µM) of Empagliflozin is appropriate for the whole experiments, due to the clinic drug blood plasma concentration wasn't mentioned by the authors.
Author Response
Responses to the Editor and the Reviewers
We thank the editor and reviewers for their valuable comments towards the improvement of the manuscript. We have modified the manuscript in accordance with your suggestions and consider it significantly improved for publication.
Reviewer 4
This study by Ahasanul Hasan and Raquibul Hasan utilized Pressure Myography method and pharmacology tools to investigate the mechanism of Empagliflozin reduce systemic blood pressure. All experiments were well preformed, but the whole experimental design is not perfect! I have some recommendations to make this study strong.
- Thinking about to do either one of the following experiments:
Comment 1a. To get the proof of SGLT2 protein expressed in the vascular smooth muscle cells from rat mesenteric arteries (if the action of Empagliflozin on Kv channels through SGLT2).
Response: We thank the reviewer for this valuable comment. In this study we demonstrated that the vasodilatory action of empagliflozin in resistance-size rat mesenteric arteries is due to the activation of KV channels. Such action of empagliflozin on Kv channels is unlikely to be mediated through SGLT-2. Although we have not measured SGLT-2 expression in arterial smooth muscle cells, which is beyond the scope of our study, previous studies were conducted to determine the tissue distribution of SGLT-2 both in human and rodents. Chen and colleagues (Chen et. al., 2010) showed that SGLT-2 is highly kidney specific by analyzing 72 normal tissues from three different individuals. In mice, renal localization of SGLT-2 protein was confirmed (Sabolic et. al., 2012), similar to that in rats. Extrarenal SGLT-2 mRNA expression was not detected. To our knowledge, no previous study has reported the expression of SGLT-2 in vascular smooth muscle cells (Alshnbari et. al., 2020). The Human Protein Atlas (HPA), which is focused on mapping all human proteins in cells, tissues and organs using integrated approaches, reported no expression of SGLT-2 in smooth muscle cells (Human Protein Atlas, 2021). We hope this addresses the concern raised by the reviewer.
Chen J, Williams S, Ho S, Loraine H, Hagan D, Whaley JM, Feder JN. Quantitative PCR tissue expression profiling of the human SGLT2 gene and related family members. Diabetes Ther. 2010 Dec;1(2):57-92. doi: 10.1007/s13300-010-0006-4. Epub 2010 Dec 17. PMID: 22127746; PMCID: PMC3138482.
Sabolic, I.; Vrhovac, I.; Eror, D. B.; Gerasimova, M.; Rose, M.; Breljak, D.; Ljubojevic, M.; Brzica, H.; Sebastiani, A.; Thal, S. C.; Sauvant, C.; Kipp, H.; Vallon, V.; Koepsell, H. Expression of Na+-D-glucose cotransporter SGLT2 in rodents is kidney-specific and exhibits sex and species differences. Am J Physiol Cell Physiol. 2012, 302(8), C1174-88. DOI: 10.1152/ajpcell.00450.2011.
Alshnbari, A.S.; Millar, S.A.; O’Sullivan, S.E. et al. Effect of Sodium-Glucose Cotransporter-2 Inhibitors on Endothelial Function: A Systematic Review of Preclinical Studies. Diabetes Ther . 2020, 11, 1947–1963. DOI: 10.1007/s13300-020-00885-z
Human Protein Atlas (2021, September 10). SLC5A2: https://www.proteinatlas.org/ENSG00000140675-SLC5A2/tissue/smooth+muscle
Comment 1b. To see the effects of Empagliflozin on Kv channels (patch clamp electrophysiology recording), utilizing exogenous expression system to over express Kv1.5 and Kv7 separately (if the action of Empagliflozin directly on Kv channels).
Response: We thank the reviewer for this valuable comment. Our study demonstrates the involvement of two KV channels in empagliflozin-induced vasodilation. Since exogenous KV expression does not adequately mimic arterial smooth muscle cell physiology, we believe that the suggested experiment would not add any value, and may lead to spurious conclusions. Our study suggests that KV channel activation is the major mechanism for mesenteric artery vasodilation. However, as empagliflozin relaxed 30K-constricted mesenteric arteries, it is possible that other mechanisms, in addition to KV channels, are also activated by this drug to induce vasodilation. Therefore, it is beyond the scope of this study to pinpoint all possible mechanisms.
Comment 2. Please include original traces for Fig. 1C, Fig. 1F, and Fig. 2.
Response: Thank your for the suggestion. We believe this is a misunderstanding about Fig 1C. The corresponding original trace is Fig. 1A. We have made an additional figure, which is Fig. 2 in the revised manuscript. This new figure shows the original traces requested and corresponding mean data.
Comment 3. Please clearly label the Y axis unit (percentage or µm) for Fig. 1B
Response: We have corrected all minor errors and typos in the revised manuscript.
Comment 4. For the Fig. 3B, please explain why the vasodilation increased dramatically after co-application of EMPA and Indomethacin.
Response: As the reviewer rightly noted, it was interesting to see a dramatic change in empagliflozin-induced vasodilation after the application of indomethacin. Although this is beyond the scope of this study, one possible explanation could be indomethacin-mediated inhibition of other COX-derived products such as PGH2 and TXA2, which are vasoconstrictors.
Comment 5. For the Fig. 6B, please explain why the vasodilation increased dramatically after co-application of EMPA and Paxilline, as well as EMPA and Glibenclamide.
Response: Thanks for this comment. Indeed, we were also intrigued by such observation. Note that, we performed all mechanistic studies in PE-constricted arteries. In the smooth muscle cells of PE-constricted arteries (that had high intracellular Ca2+ due to the release and influx of Ca2+), inhibition of KV channels could block K+ efflux, leading to further depolarization and subsequent activation of Ca2+-sensitive K+ channels like BKca, IKca and SKca channels, which could lead to arterial vasodilation. Blocking of one Ca2+-sensitive K+ channel at a time may still sustain such vasodilation via this mechanism. In a separate study (unpublished), we found that application of TEA that blocks all K+ channels does not cause such vasodilation. This may suggest that TEA prevents the activation of all Ca2+-sensitive K+ channels, and therefore, vasodilation. This could be a plausible explanation for the observed vasodilation. However, other mechanisms may also operate.
Comment 6. For the experiment of Fig. 1A, I am wondering if the vasodilation response will be reversed when the 100 µM Empagliflozin was washed away.
Response: Yes, the vasodilation is fully reversed after 5 minutes of wash with 6K PSS.
Comment 7. I don’t know whether using such high concentration (100 µM) of Empagliflozin is appropriate for the whole experiments, due to the clinic drug blood plasma concentration wasn't mentioned by the authors.
Response: Thanks for raising this point. Although we used 100 µM for the mechanistic studies, we observed concentration-dependent vasodilation in myogenic, PE- and 30K-constricted arteries. Empagliflozin is prescribed up to 100 mg/day. Empagliflozin is absorbed rapidly and reaches a peak plasma concentration of 2.75µΜ within 3 hours after administration of 100 mg (Heise et al., 2013). We agree that the concentration of empagliflozin used was higher than that used clinically but this is only for the mechanistic study. A lower empagliflozin concentration, 1μM that falls within the therapeutic range also evoked strong vasodilation when compared to 100µM response (34 µm Vs. 60 µm) (Fig. 1B). Therefore, 1μM and smaller concentrations are likely to produce vasodilation that could influence blood flow and pressure. Furthermore, overmedication or abuse of empagliflozin can raise the blood concentration of empagliflozin. Therefore, our results should be fully considered when prescribing empagliflozin to patients with cardiovascular diseases such as hypotension (Seo et. a. 2020).
Heise, T.; Seman, L., Macha, S.; Jones, P.; Marquart, A.; Pinnetti, S., Woerle, H. J.; Dugi, K. Safety, tolerability, pharmacokinetics, and pharmacodynamics of multiple rising doses of empagliflozin in patients with type 2 diabetes mellitus. Diabet. Ther. 2013, 4, 331–345. DOI: 10.1007/s13300-013-0030-2
Seo, M. S.; Jung, H. S.; An, J. R.; Kang, M.; Heo, R.; Li, H.; Han, E. T.; Yang, S. R.; Cho, E. H.; Bae, Y. M.; Park, W. S. Empagliflozin dilates the rabbit aorta by activating PKG and voltage-dependent K(+) channels. Toxicol Appl Pharmacol. 2020, 403, 115153. DOI: 10.1016/j.taap.2020.115153
Round 2
Reviewer 1 Report
It'll be nice to add the study data on diabetic models, otherwise I have no further comment.
Author Response
Thank you for your valuable comments. Our future study will investigate those further.
Reviewer 3 Report
No additional concerns.
Author Response
Thank you for your valuable comments.
Reviewer 4 Report
I have no further comments.
Author Response
Thank you for your valuable comments.